# ECA-VFog: An efficient certificateless authentication scheme for 5G-assisted vehicular fog computing

**Abdulwahab Ali Almazroi**[1][☯]*, **Eman A. Aldhahri**[2][☯], **Mahmood A. Al-Shareeda**[ID][3][☯]*, **Selvakumar Manickam**[3][☯]

**1** Department of Information Technology, College of Computing and Information Technology at Khulais, University of Jeddah, Jeddah, Saudi Arabia, **2** Department of Computer Science and Artificial Intelligence, College of Computer Science and Engineering, University of Jeddah, Jeddah, Saudi Arabia, **3** National Advanced IPv6 Centre (NAv6), Universiti Sains Malaysia, Gelugor, Penang, Malaysia

☯ These authors contributed equally to this work.
* aalmazroi@uj.edu.sa (AAA); alshareeda022@usm.my (MAAS)

**Data Availability Statement:** All relevant data are within the paper.

## Abstract

Fifth-generation (5G)-enabled vehicular fog computing technologies have always been at the forefront of innovation because they support smart transport like the sharing of traffic data and cooperative processing in the urban fabric. Nevertheless, the most important factors limiting progress are concerns over message protection and safety. To cope with these challenges, several scholars have proposed certificateless authentication schemes with pseudonyms and traceability. These schemes avoid complicated management of certificate and escrow of key in the public key infrastructure-based approaches in the identity-based approaches, respectively. Nevertheless, problems such as high communication costs, security holes, and computational complexity still exist. Therefore, this paper proposes an efficient certificateless authentication called the ECA-VFog scheme for fog computing with 5G-assisted vehicular systems. The proposed ECA-VFog scheme applied efficient operations based on elliptic curve cryptography that is supported by a fog server through a 5G-base station. This work conducts a safety analysis of the security designs to analysis the viability and value of the proposed ECA-VFog scheme. In the performance ovulation section, the computation costs for signing and verification process are 2.3539 ms and 1.5752 ms, respectively. While, the communication costs and energy consumption overhead of the ECA-VFog are 124 bytes and 25.610432 mJ, respectively. Moreover, comparing the ECA-VFog scheme to other existing schemes, the performance estimation reveals that it is more cost-effective with regard to computation cost, communication cost, and energy consumption.

## 1 Introduction

Technologies related to automobiles have consistently ranked among the most promising areas of research and development. Improvements in human well-being have a knock-on effect on automotive engineering [1–3]. Nowadays vehicle networks are in the spotlight for a variety

**Funding:** The authors extend their appreciation to the Deputyship for Research & Innovation, Ministry of Education in Saudi Arabia for funding this research work through project number MoE-IF-UJ-22-04100409-3.

**Competing interests:** The authors have declared that no competing interests exist.

of reasons, including urban traffic congestion and road accidents. Many crucial traffic issues are communicated to users via vehicle networks, including speed alerts, cornering, road status information, road conditions, intersection warnings, and pedestrian crossing alerts [4–6].

Several countries' transportation systems have recently implemented widespread deployments of 5G technology, vehicle networks, and fog computing to enhance driver safety and better handle increasingly chaotic traffic patterns [7–9]. Intelligent transportation systems (ITS) collect, process, and disseminate traffic data in the context of networked cars through the use of wireless devices installed in vehicles (called onboard units, or OBUs) [10–12].

Because traffic-related messages travel through a wireless channel, they are vulnerable to eavesdropping, tampering, replaying, and deletion by hostile actors [13, 14]. So, vehicular ad hoc networks (VANETs) need to address privacy and security concerns before they can be used in real-world applications. Several works have been presented over the past few years concerning authentication schemes for vehicular communication. These works range from public key infrastructure (PKI) approaches [15–19] and identity (ID) approaches [5, 11, 20–25]. In PKI-based approaches, the trusted authority forces a huge number of security keys and relevant certificates onto the vehicles to ensure the security of users' private data. However, certificate management complexity is a major drawback of these schemes. While, in the identity (ID)-based approaches, the message is signed using the transmitter's secret key, and the receiver's public key is utilised to verify the signature. But, key escrow is a major flaw in these schemes.

Thus, to resolve these issues in PKI-based and ID-based approaches, several scholars have suggested certificateless authentication approaches in order to avoid complicated management of certificate and escrow of key, respectively. However, challenges such as expensive communication, insecure systems, and complicated processing remain. This research, therefore, presents an effective certificateless authentication mechanism for vehicle fog computing over 5G networks; we term it ECA-VFog. The proposed ECA-VFog technique utilized a 5G-base station-supported fog server for its elliptic curve cryptography-based efficient operations. The main lists of the contribution of this work are as follows.

- This paper suggests an efficient certificateless authentication called ECA-VFog scheme for fog computing with a 5G-assisted vehicular system. The proposed ECA-VFog scheme applied efficient operations based on elliptic curve cryptography that is supported by a fog server through a 5G-base station.

- The innovative of the proposal is that the fog severer receives partial pseudonym-ID and partial private key from key generation center (KGC) for the signature verification process.

- The ECA-VFog scheme avoids complicated management of certificate and escrow of key in the public key infrastructure-based studies and in the identity-based approaches, respectively.

- Security evaluation shows that the proposed ECA-VFog scheme fulfills security requirements (data authenticity and integrity, pseudonym identity, traceability, unlinkability, location privacy, non-repudiation) and resists security attacks (forgery, modified messages, replay, and man-in-the-middle) for vehicular fog computing based on 5G technology.

- The evaluation of the ECA-VFog scheme's performance shows that it is more efficient than existing schemes with regards to computation cost, communication overhead, and energy consumption.

Here's how the remainder of the work is laid out: Section 2 shows some relevant work. Section 3 lists the architecture model, security design and cyber-attacks, and operation-based mathematical tool of our ECA-VFog. In Section 4, our ECA-VFog and it's implementation phases are given. Informal and formal analysis are lsited in Section 5. The evaluation of performance is analyzed in Section 6. This study is concluded and summarized in Section 7.

## 2 Literature review

This section reviews some relevant work that proposed authentication schemes for vehicular communication. We classify these schemes based on approaches used to secure messages. These taxonomies are public key infrastructure (PKI)-based, identity (ID)-based, and certificateless authentication approaches.

### 2.1 PKI-based approaches

Many PKI-based approaches [15–19] have been proposed to secure vehicular systems. These schemes are reviewed as follows. Sakhreliya et al. [15] presented the PKI-SC system, which combines the best of both worlds by integrating the MAC technique into the standard PKI certificate process. The MAC and ECDSA algorithms are deployed on the nodes in order to make a fair comparison among the PKI and PKI-SC systems, and the packet size is utilized to evaluate the time it takes for each system's communications to complete a given task. Utilizing the ideas of Bayesian Coalition Game (BCG) and Learning Automata (LA), Kumar et al. [16] developed an effective decentralized PKI. Los Angeles was supposed to be the game's participants, who work together to share information. In their solution, a coalition of dynamic between the users is created utilizing encryption of symmetric key and message authentication based on hash to protect the privacy and authenticity of the exchanged information. Jiang et al. [17] designed a PKI-based pseudonym authentication scheme by establishing secure session keys and providing the method of disclosing malicious vehicular to construct a timestamp signature. Zhang et al. [18] proved a PKI identity management based on blockchain and model of authentication, which makes use of smart contracts to lessen the load on TRA from handling the entire digital certificate life cycle alone. Moussaoui et al. [19] suggested a decentralized system for pseudonym management in vehicular communication. To carry out the many anonymity-related tasks, Moussaoui et al. [19] employed blockchain technology that calls for two separate blockchains: one for registering aliases and another for deleting them. Nevertheless, the main disadvantage of these schemes is complicated management of certificate.

### 2.2 ID-based approaches

Many ID-based approaches [5, 11, 20–25] have been suggested to address limitations on PKI-based schemes. These schemes are reviewed as follows. In order to achieve the goals of confidentiality, anonymity, and security in a VANET, Alazzawi et al. [20] suggested a novel ID-based approach. In the event that the roadside unit (RSU) is compromised, the proposed approach used a pseudonym during the joining procedure to conceal the true identity. For secure vehicle-to-vehicle (V2V) data exchange, Ali et al. [21] suggested applying Elliptic Curve Cryptography (ECC) and general hash functions to create ID-based approaches by using the batch signature investigation mode, a huge volume of data can be authenticated simultaneously. In order to secure V2V communications over vehicular systems, Bansal et al. [22] offered an Identity-based authentication method that makes use of both ID and ECC. Through efficient V2V communications, the approach guaranteed source verification, data integrity, non-repudiation, and vehicle anonymity. Mohammed et al. [23] designed a pseudonym

authentication based on fog computing to minimize the performance efficiency in 5G-assisted vehicular systems. The FC-PA study performs only one operation of ECC scalar multiplication to check data. By using fog computing technology, Al-Mekhlafi et al. [5] introduced an authentication scheme for 5G-assisted vehicle systems. A fog server computes and stores a unique selecting of public anonymity identities and signature keys for each legal component. For fog computing with 5G-assisted vehicular systems, Mohammed et al. [11] proposed a pseudonym authentication technique. Under their works, a fog server generates a temporary secret key for each participating vehicle to use for validating digital signatures. To counteract potential side-channel attacks and slow down the system, Alshudukhi et al. [24] constructed an authentication technique with supporting a privacy factors. In addition, the TPD regularly and often updates its most important data in an effort to thwart side-channel attacks. Bayat et al. [25] suggested an innovative and effective authentication method for vehicular communication by enabling vehicles to authenticate each other without the usual restrictions imposed by the necessity for a designated group of signers, an active network of Road Side Units (RSUs), a secret key, or other similar safeguards. However, the major disadvantage of these schemes is escrow of key.

## 2.3 Certificateless authentication approaches

To cope with these issues, massive certificateless authentication studies with pseudonyms and traceability have been suggested. These studies avoid complicated certificate management and key escrow in the PKI-assisted studies and the ID-assisted studies, receptively. These schemes are as below. Wang et al. [26] construed a privacy factors scheme by adopting a full aggregation approach for reducing resources in terms of bandwidth and computation. Xu et al. [27] constructed certificateless fixed checker proxy signature using unmanned aerial vehicles (UAVs) to address privacy and security concerns in smart city systems. Ming et al. [28] suggested an efficient certificateless authentication scheme by achieving a security-enhanced solution and addressing massive communication overhead, security vulnerability, and computational complexity. Tan et al. [29] proposed a certificateless UAV group verification approach in order to achieve security communication in infrastructure-less internet of vehicle (IoV). Zhou et al. [30] introduced a secure ECC scheme by utilizing key agreement and a three-party authentication scheme in medical IoT. Rajasekaran et al. [31] proposed a secure ECC method that supports batch verification and mutual authentication for online learning in Industry 4.0. Zhou et al. [32] introduced a security-enhanced solution to combat a forgery attack and satisfy a trade-off between efficiency and safety in vehicular communication. Liang et al. [33] evaluated the safety of a certificateless aggregate signature for vehicular communication, focusing on the preservation of privacy under certain conditions. The investigation reveals that it is suffering from forgery attacks. Thus, Liang et al. [33] proposed a better strategy to cope with the security flaw.

## 3 Background

This section demonstrates the architecture model of the proposal ECA-VFog scheme in terms of the five components used. Then, we list the security design and cyber-attacks that should be resisted in this paper. Finally, the operation-based mathematical tool used to sign and verify messages is also provided.

### 3.1 Architecture model

Our architecture model has five parts, as depicted in Fig 1: A tracing authority (TRA), a key generation center (KGC), a 5G-base station (5G-BS), a fog server (FS), and onboard units

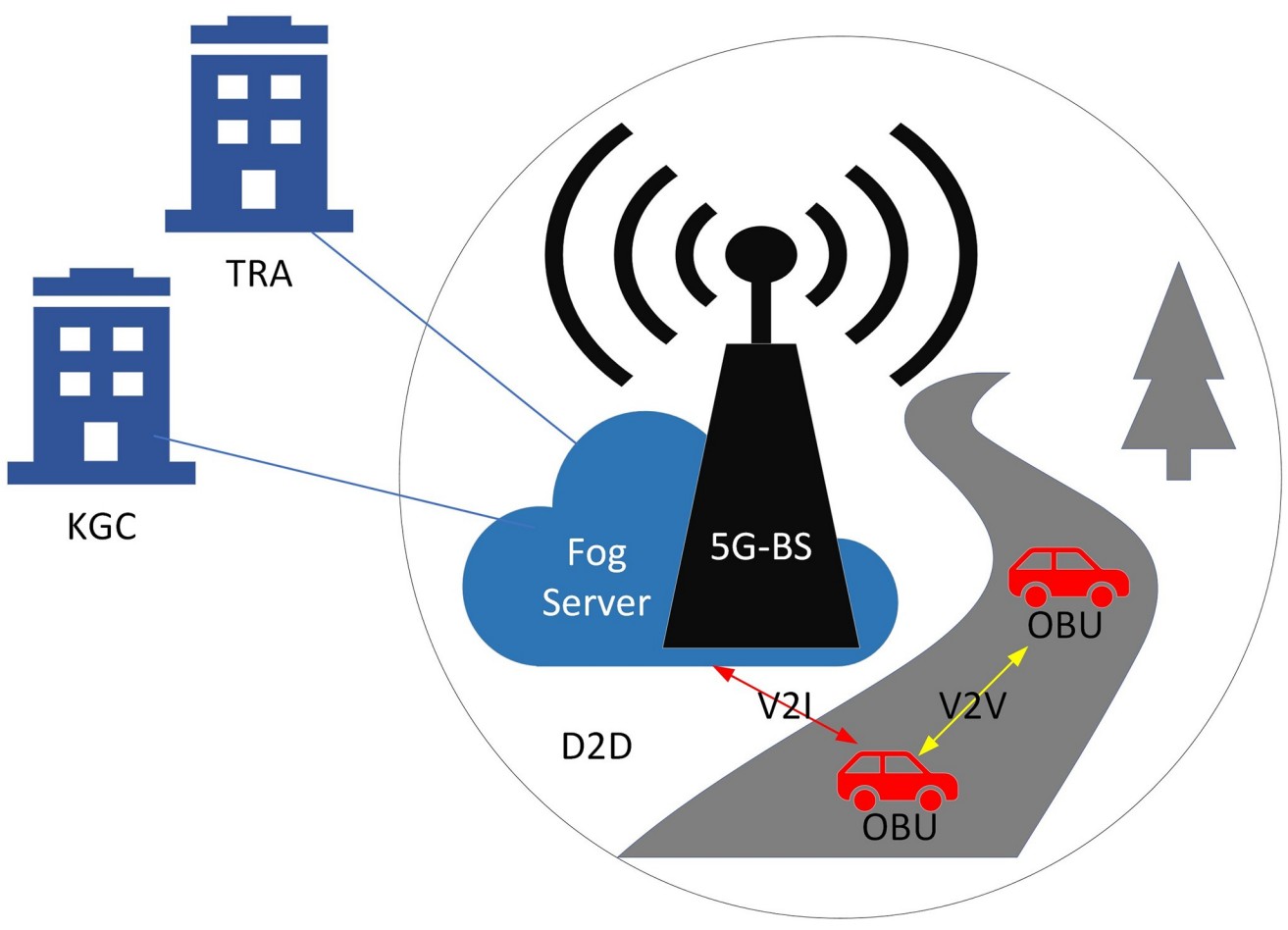

**Fig 1. Architecture model for the ECA-VFog scheme.**

(OBUs). Since 5G-BS doesn't be able to compute and storage any security parameters. Therefore, we don't care the assumption of this proposed in term trust or not trued. While, this paper assumes fog server is not trusted, therefore, the KGC preload partial public key and partial signature to communication with vehicles. Finally, the TRA and KGC are fully trusted in this system model to generate security parameters. The following are some of the roles that these parts play.

- Tracing Authority (TRA): Vehicle registration in the vehicular system falls under the purview of the TRA, a central authority in the field. Its other duty is to provide the KGC with a means of partial anonymity. When a malicious event is detected, only the TRA will reveal the true identities of the vehicles and fog servers. Contacting TRA on a regular basis allows vehicles and fog servers to keep their credentials up to date. So they are still part of vehicular communication. If the TRA has previously identified malicious behavior from a user (vehicle/fog server), the TRA will refrain from performing identity updates for that user.

- Key Generation Center (KGC): As a credible source, KGC is an asset. It is compatible with TRA and can produce vehicle and RSU partial private keys (PPK). Partial key generation in related schemes is no longer hampered by the need to escrow keys. Related schemes

incorporate KGC, which is absent from identity-based authentication schemes. It helps TRA establish the anonymous identities of vehicles and fog servers, too. Only TRA knows the true identity, and KGC can only get a glimpse of this fake one.

- 5G-Base Station (5G-BS): The 5G-BSs are stationary base stations set up by the side of the road. Its only use is as a bridge between vehicles, fog servers, and TRA, and it lacks both computing and storage capabilities. This is because it can accommodate a wide variety of device-to-device (D2D) communication standards. Because 5G-BSs are hardware, they are immune to attacks.

- Fog Server: Fog server is the roadside infrastructure that enables vehicle-to-infrastructure (V2I) communication and can also realize inter-infrastructure (I2I) communication. FSs can simultaneously relay multiple messages collected from vehicles. FSs are stationed in various areas behind 5G-BS, and passing vehicles are made aware of their location. By sharing information, it can also boost circulation in the area covered by 5G-BS.

- Onboard Units (OBUs): OBUs are installed in car. They employ vehicle-to-vehicle (V2V) to talk to one another and V2I to talk to the fog server. Traffic and signature-related messages are generated by vehicles and sent to other vehicles or fog servers. There is a tamper-proof device (TPD) in the vehicles. The data on this device is strictly private.

## 3.2 Security design

The safety of the vehicular network is compromised by cyber attacks, which can even result in human casualties. The following sections detail the minimum standards for security against cyber attacks and unauthorized access that the ECA-VFog scheme in the 5G-enabled vehicular fog computing must meet.

- Message authenticity and safety: To emphasize the integrity of the received data, the receiving component must first confirm that the sending vehicle is also registered in the vehicular communication.

- Pseudonym and Traceability: Due to security concerns, the true identities of the vehicles and fog servers must be concealed, so they instead adopt aliases. The pseudonym of vehicles or fog servers does not make them immune to detection when they are engaged in criminal activity. The TRA reveals their true identities here. This guarantees that the vehicles can be tracked at any time.

- Un-linkability: An adversary must be unable to connect identical vehicle or fog server-generated signatures and messages. A new identity must be used for each transmission even if the vehicle and fog server sends the message anonymously. The message and the sender's identity can be linked if the sender's pseudonym information is not altered.

- Non-repudiation: The content of transmissions from vehicles and fog servers is their own responsibility. They should be unable to deny it even if they send messages using fake names and signatures.

- Location Privacy: Protecting the confidentiality of vehicle locations is crucial for their safety. There are a number of means that attackers can use to track down vehicles. To protect their users' location privacy from attackers, vehicles use pseudonym identities rather than their real names, and these identities are randomly generated for each message sent.

- Forgery Attack: An adversary posing as a network user (vehicle/fog server) can send messages to other tools and fog servers in vehicular communication.

- Various cyber attacks: Cyber attacks such as replay attacks, man-in-the-middle attacks, modified messages, etc., are extremely common in VANETs.

### 3.3 Elliptic Curve Cryptography (ECC)

Since it is determined on the finite field $F_p$, ECC is an encryption technique of public-key. The equation $y^2 = x^3 + ax + b \pmod{p}$, where $D$ is a massive enough prime value and $a$, $b$, characterizes the elliptic curve $E$ based non-singular over a finite field $F_p$. It is very complex to reach $s$ in the equation $Q = s \cdot P$ if $D$ and $R$ are known, making ECC secure thanks to the issue of the elliptic curve discrete logarithm (ECDL).

ECC is used for operations such as subtraction, doubling, point addition, and scalar multiplication. Let's pretend two points of $D$ and $R$ in space over E/F$p$. Definitions for scalar multiplication ($k D = D + D + \ldots + D$ (k times)), the addition of point ($D + R = R$), doubling of point ($D + D = 2D$), subtraction of point ($D R = D + (R) = R$), and addition of point ($D + D = 2D$) are all possible.

## 4 The proposed ECA-VFog scheme

The proposed ECA-VFog scheme includes phases, Setup, Registration phase, GenPPID, GenPPK phases, GenCLSig and CLSigVerify, as shown in Fig 2. Unlike the related works, in the Setup phase, both TRA and KGC issue system parameters based on the elliptic curves and broadcast them to register vehicles and fog servers through the registration stage. While KGC is responsible for creating the partial pseudonym-ID $PPID_V$ and partial private key $PPK$ for

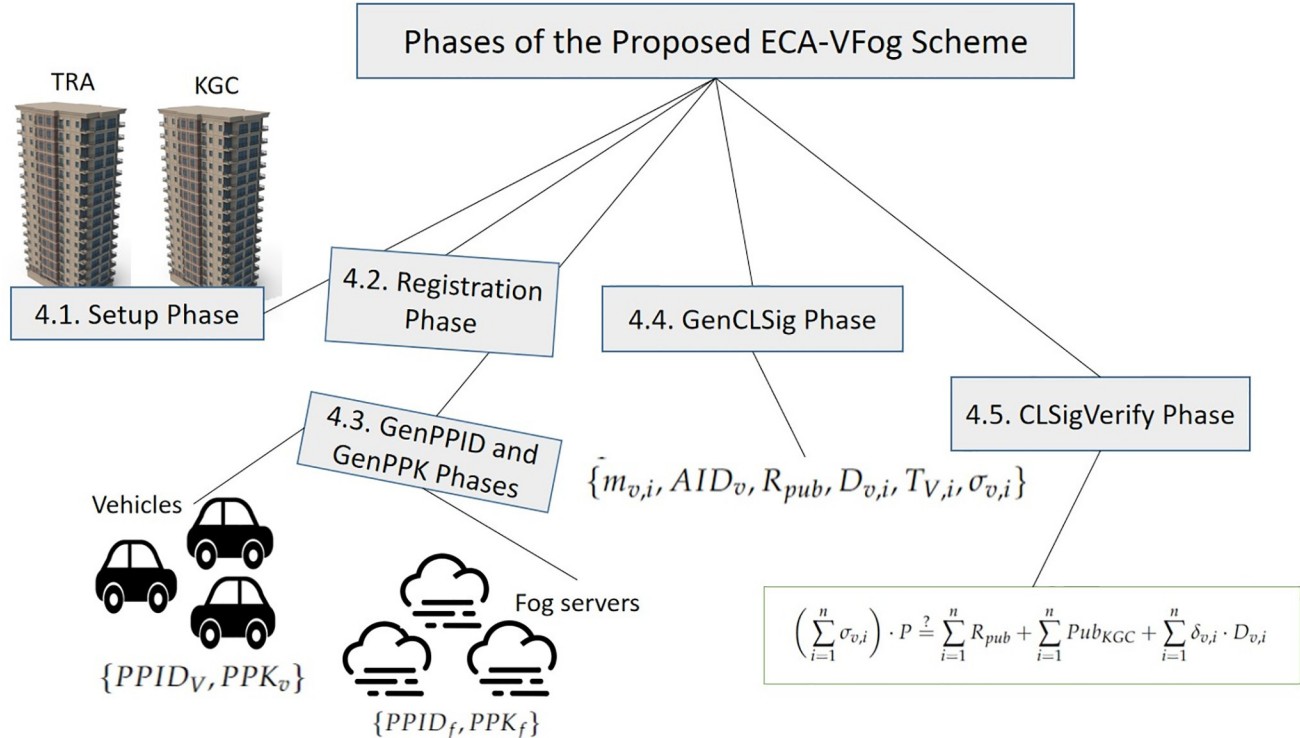

**Fig 2. Proposed ECA-VFog scheme.**

**Table 1. Notation and their definition.**

| Notation | Definition |
|---|---|
| ECA | Efficient Certificateless Authentication |
| TRA | Tracing Authority |
| KGC | Key Generation Center |
| PKI | Public Key Infrastructure |
| ID | Identity |
| $F_p$ | Finite field |
| $a, b$ | Major prime values |
| $P$ | Generator of elliptic curve |
| $h_1, h_2, h_3, h_4$ | General secure hash functions |
| $s, Pub_{TRA}$ | TRA's key of secret and the relevant key of public, respectively |
| $x, Pub_{KGC}$ | KGC's key of secret and the relevant key of public, respectively |
| $Ps_v, VID_i$ | Vehicle $V_i$'s pseudonym and original identity, respectively |
| $Ps_f, FID_i$ | Fog server $F_i$'s pseudonym and original identity, respectively |
| $PPID_V, PPK_v$ | Vehicle $V_i$'s partial pseudonym-ID and partial private key, respectively |
| $PPID_F, PPK_F$ | Fog server $F_i$'s partial pseudonym-ID and partial private key, respectively |
| $\chi, \lambda$ | A secret key of vehicle and fog server, respectively |
| $Pri_v$ | Vehicle secret key of $r_i$ |
| $AID_v$ | Anonymous identity of vehicle |
| $Pri_v, R_{pub}$ | Private key and public key of vehicle |
| $\sigma_{v,i}$ | Signature of the $i^{th}$ vehicle |
| $T_i$ | The timestamp of the system. |
| $\|$ | The concatenation operation. |
| $\oplus$ | The exclusive-OR operation |

vehicles and fog servers to maintain the original identity during GenPPID and GenPPK phases. According to the GenCLSig phase, the transmitter signs the data by generating a signature and secret parameters, while the receiver will check the validity and originality of the data during the CLSigVerify stage. These phases are described in detail as follows. Table 1 lists notations and their definition.

### 4.1 Setup phase

In order to design a secure and effective ECA-VFog scheme, the most crucial part of setting up a system is choosing its parameters. The steps for this stage are outlined below.

- Both the TRA and the KGC agree that for any finite field, $F_p$, an elliptic curve $E(a, b)$ exists if and only if $p$ is a sufficiently big prime amount and $a$, $b$ are fixed integers less than $p$. The expression $y^2 = x^3 + ax + b \pmod{p}$ defines the E(a, b).

- Both the TRA and the KGC pick values for $a$ and $b$ and verify that $4a^3 + 27b^2 \neq 0. \pmod{p}$. If the condition of equality is not met, $a$ and $b$ are chosen again. Then, a starting point $P$ of level $q$ is chosen.

- The four functions of general secure hash $h_{1,2,3,4}$ are selected as follows. $h_1 : G \times {0, 1}^* \rightarrow Z_q^*$, $h_2 : G \rightarrow Z_q^*$, $h_3 : G \rightarrow Z_q^*$ and $h_4 : {0, 1}^* \times {0, 1}^* \times G \times G \times {0, 1}^* \rightarrow Z_q^*$.

- The TRA picks a secret key $s \in Z_q^*$ and calculates the relevant key of public $Pub_{TRA} = s \cdot P$.

- The KGC picks a secret key $x \in Z_q^*$ and issues the relevant key of public $Pub_{KGC} = x \cdot P$.

- Finally, both the TRA and the KGC broadcast system initial parameters {$E$, $a$, $b$, $P$, $p$, $q$, $Pub_{TRA}$, $Pub_{KGC}$, $h_3$, $h_4$} in the vehicular communication.

Note that only the TRA and KGC make use of the $h_1$ and $h_2$ hash functions; as a result, they are not made public. Both the TRA and KGC use hash functions $h_1$ and $h_2$ to create a pseudonymous ID. The TRA then uses the $h_1$ and $h_2$ hash functions to find out who the cars and fog servers really are.

## 4.2 Registration phase

In this phase, fog servers and vehicles are given unique identifiers that can be used to register with the TRA. Here's how this step is carried out:

- The TRA calculates $Ps_v = VID_i \oplus h_1(Pub_{TRA}||S)$ utilising vehicle $V_i$'s original identity $VID_i$ and its private key $s$. $Ps_v$ is a constraint amount for the vehicle $V_i$ and its original identity $VID_i$ can only be disclosed by the TRA.

- Likewise, the TRA executes the same procedure for the fog server $F_i$. The TRA calculates $Ps_f = FID_i \oplus h_1(Pub_{TRA}||S)$ utilizing fog server $F_i$'s original identity $FID_i$, and its private key $S$. $Ps_f$ is a constraint amount for the fog server and its original identity ($FID_i$) can only be disclosed by the TRA.

- The TRA loads the values of $Ps_v$ and $Ps_f$ in the TPD of the vehicle $V_i$ and fog server $F_i$, respectively.

## 4.3 GenPPID and GenPPK phases

The privacy and security of vehicular communication depend on the vehicles' and fog servers' ability to maintain their true identities at all times. KGC generates partial pseudonym-ID $PPID$ and partial private key $PPK$ at this phase. Vehicle's partial pseudonym-ID $PPID_V$ and fog server's partial pseudonym-ID $PPID_F$ are known only to TRA to satisfy traceability requirements. The following procedures are carried out during this stage for vehicles and fog servers.

### 4.3.1 Vehicles.

- The vehicle $V_i$ inputs the value of $Ps_v$ to the KGC.

- The KGC verifies $Ps_v$ in vehicle database sent by TRA. Vehicles that have received fines or that are not properly registered are not included in this database. The KGC will continue its calculations if the value $Ps_v$ is present in the database; otherwise, it will exit.

- The vehicle $V_i$'s partial pseudonym-ID $PPID_V = Ps_v \oplus h_2(x \cdot Pub_{TRA})$ is calculated by utilizing its secret key $x$.

- The KGC chooses a secret key $\chi \in Z_q^*$.

- The KGC calculates vehicle $V_i$'s partial private key $PPK_v = x + \chi \cdot h_1(PPID_V||x||\chi Pub_{KGC})$.

- The KGC transmits parameters {$PPID_V$, $PPK_v$} to the vehicle $V_i$ via a secure channel.

### 4.3.2 Fog servers.

- The fog server $F_i$ inputs the value of $Ps_f$ to the KGC.

- The KGC verifies $Ps_f$ in fog server database sent by TRA. Fog servers that have received fines or that are not properly registered are not included in this database. The KGC will continue its calculations if the value $Ps_f$ is present in the database; otherwise, it will exit.

- The fog server $F_i$'s partial pseudonym-ID $PPID_f = Ps_f \oplus h_2(x \cdot Pub_{TRA})$ is calculated by utilizing its secret key $x$.

- The KGC chooses a secret key $\lambda \in Z_q^*$.

- The KGC calculates vehicle $V_i$'s partial private key $PPK_f = x + \lambda \cdot h_1(PPID_f||x||\lambda Pub_{KGC})$.

- The KGC transmits parameters $\{PPID_f, PPK_f\}$ to the fog server $F_i$ through a secure channel.

## 4.4 GenCLSig phase

Before sending a message, the vehicle signs it to ensure its safety. Therefore, the vehicle that received the message checks its veracity. After receiving the PPK and PPID calculated by the KGC, the vehicle is able to send messages to other vehicles and fog servers. As a result, the proposed ECA-VFog scheme eliminates the need for vehicles and fog servers to continuously communicate with the KGC and the TRA. The following steps are taken during this stage.

- The vehicle $v_i$ randomly select secret key $r_i \in Z_q^*$.

- The vehicle $v_i$ computes its vehicle secret key $Pri_v = PPK_v + r_i$ utilising the secret key $r_i$ and $PPK_v$.

- The vehicle $v_i$ computes $R_{p,i} = Pri_v \cdot P$.

- The vehicle $v_i$ computes its public key $R_{pub} = R_{p,i} - Pub_{KGC}$.

- The vehicle $v_i$ computes its anonymous identity $AID_v = PPID_v \oplus h_3(r_i \cdot Pub_{TRA})$ utilising $r_i$ and partial anonymous identity $PPID_v$ for each message.

- The vehicle $v_i$ computes $D_{v,i} = r_i \cdot P$ and $\delta_{v,i} = h_4(m_{v,i}||AID_v||R_{pub}||D_{v,i}||T_{V,i})$, where $m_{v,i}$ is exchanged message $T_{V,i}$ is freshness timestamp, and anonymous identity $AID_v$.

- The vehicle $v_i$ issues signature $\sigma_{v,i} = Pri_v + r_i \cdot \delta_{v,i}$ (mod q).

- Finally, the vehicle $v_i$ broadcasts the message-parameters $\{m_{v,i}, AID_v, R_{pub}, D_{v,i}, T_{V,i}, \sigma_{v,i}\}$ to other vehicles and fog servers.

## 4.5 CLSigVerify phase

In this phase, the vehicle or fog server that received the message performs an authentication and integrity check to decide whether or not to accept the message. This step is carried out as follows for a verification process.

- The verifier controls the timestamp $T_{V,i}$ of the data obtained at period $T$. The message is declined if $T - T_{V,i} > \Delta T$. If not, it advances to the next level. This means that messages that have already expired are deleted without being read.

- The verifier computes $\delta_{v,i} = h_4(m_{v,i}||AID_v||R_{pub}||D_{v,i}||T_{V,i})$ and tests whether it fulfills (1) utilizing the message-tuples $\{m_{v,i}, AID_v, R_{pub}, D_{v,i}, T_{V,i}, \sigma_{v,i}\}$ for a single data. If not, the message

$m_{v,i}$ is discarded.

$$\sigma_{v,i} \cdot P \overset{?}{=}$$

$$(Pri_v + r_i \cdot \delta_{v,i}) \cdot P$$

$$(PPK_v + r_i + r_i \cdot \delta_{v,i}) \cdot P$$

$$(Pri_v + r_i \cdot \delta_{v,i}) \cdot P \tag{1}$$

$$Pri_v \cdot P + \cdot \delta_{v,i} r_i \cdot P$$

$$R_{p,i} + \delta_{v,i} \cdot D_{v,i}$$

$$R_{pub} + Pub_{KGC} + \delta_{v,i} \cdot D_{v,i}$$

As a result, (1) can be demonstrated. It is determined during the verification process by employing the message and previously published parameters. If any of these values are altered, the signature can no longer be validated. Meanwhile, the proposed offers batch authentication to raise efficiency in traffic. The verifier checks Eq 2 once receiving multiple messages.

$$\left(\sum_{i}^{n} \sigma_{v,i}\right) \cdot P \overset{?}{=} \sum^{n} R_{pub} + \sum^{n} Pub_{KGC} + \sum^{n} \delta_{v,i} \cdot D_{v,i} \tag{2}$$

## 5 Security evaluation

This section evaluates security with regards to informal and formal analysis as follows.

### 5.1 Security analysis

- Message Authenticity and Integrity: The unit and the fog server use the message parameters $\{m_{v,i}, AID_v, R_{pub}, D_{v,i}, T_{V,i}, \sigma_{v,i}\}$ to determine if the equation $\sigma_{v,i} \cdot P \overset{?}{=} R_{pub} + Pub_{KGC} + \delta_{v,i} \cdot D_{v,i}$ holds. $\delta_{v,i} = h_4(m_{v,i}||AID_v||R_{pub}||D_{v,i}||T_{V,i})$ is computed by the receiver (vehicle/fog server) using a timestamp $T_{V,i}$, and an anonymous identity $AID_v$, $R_{pub}$, and $m_{v,i}$ are the car's public key and message, respectively. This ensures that the received message is both authentic and intact. If one variable is changed, the other cannot bring about parity. As a result, our ECA-VFog method proves the authenticity and confidentiality of communications.

- Pseudonym Identity: Pseudonym is used by vehicles and fog servers because it is necessary for security reasons to conceal their true identities [34]. Vehicles and fog servers may be able to avoid detection when they engage in illegal activity by using false identities; however, this is not always the case. Their true identities are exposed here thanks to the TRA. With the ECA-VFog scheme, the vehicle (or fog server) generates its own unique pseudonym identity with each signature. Before this happens, TRA, KGC generate a pseudonym partial identifier for the vehicle (or fog server): $Ps_v = VID_i \oplus h_1(Pub_{TRA}||S)$, $PPID_V = Ps_v \oplus h_2(x \cdot Pub_{TRA})$. To ensure privacy, TRA, and KGC issue vehicles with only partial anonymity; the vehicles then use $r_i$ and the partial pseudonym identity $PPID_V$ to determine the anonymous identity $Ps_v$ for each message. As a result, the proposed ECA-VFog scheme ensures the pseudonym identity of communications.

- Traceability: If $h_3(r_i \cdot Pub_{TRA}) = h_3(r_i \cdot s \cdot P)$, and $D_{v,i} = r_i \cdot P$, then $AID_v = PPID_v \oplus h_3(S \cdot D_{v,i})$, where $s$ is TRA's private key and $AID_v$ is the pseudonym identity, can be derived using these equations. With these numbers, the TRA can piece together the vehicle's partial anonymous identity, known as $AID_v$. In conclusion, TRA is the only method capable of revealing the identity, with $AID_v = PPID_v \oplus h_3(S \cdot D_{v,i})$. Therefore, the ECA-VFog allows for identification tracking while also protecting users' privacy (anonymity).

- Unlinkability: No two signatures or messages from the same vehicle or fog server should be linked in an attacker's mind. If the pseudonym identity is not altered between each message broadcast, it is still possible to determine the sender and recipient of the message. Each time a signature is calculated in the ECA-VFog scheme, the vehicle $AID_v$, $T_{V,i}$, $R_{pub}$ used in the calculation are chosen at random. The message parameters $\{m_{v,i}, AID_v, R_{pub}, D_{v,i}, T_{V,i}, \sigma_{v,i}\}$ that are transmitted along with the message are dynamic and can change for each transmission. As a result, the ECA-VFog scheme guarantees unlinkability.

- Location privacy: A vehicle's location and communications should be kept private through the use of a pseudonym identity. Each message in the proposed ECA-VFog scheme undergoes a pseudonym identity calculation prior to transmission. This prevents the attacker from linking any of the messages together. The proposed ECA-VFog scheme is a pseudonym and cannot be traced back to an individual. All of these characteristics have been shown above. Location secrecy is thus ensured by the ECA-VFog.

- Non-repudiation: The transmissions made by vehicles and RSUs are their own responsibility. Their true identity will be exposed if they deny sending the message. Therefore, there is no way for the sender (vehicle/fog server) to claim that it did not send the message. $AID_v = PPID_v \oplus h_2(S \cdot Pub_{KGC}) \oplus h_1(S||Pub_{TRA})$ is the TRA formula that will reveal the identity of the pseudonym person. Thus, ECA-VFog ensures that claims cannot be contested.

- Impersonation attack: The impersonation assault requires the attacker to craft a forged message $m_{v,i}$ and its the message parameters $\{m_{v,i}, AID_v, R_{pub}, D_{v,i}, T_{V,i}, \sigma_{v,i}\}$ that fulfill the equation $\sigma_{v,i} \cdot P \overset{?}{=} R_{pub} + Pub_{KGC} + \delta_{v,i} \cdot D_{v,i}$. Since this is based on ECDLP, however, it is quite challenging to implement. So, our ECA-VFog method is secure against impersonation.

- Modification attack: When the message is checked to see if it contains the equation $\sigma_{v,i} \cdot P \overset{?}{=} R_{pub} + Pub_{KGC} + \delta_{v,i} \cdot D_{v,i}$, any tampering by the attacker is immediately revealed. Since $\delta_{v,i} = h_4(m_{v,i}||AID_v||R_{pub}||D_{v,i}||T_{V,i})$, equality cannot be achieved if even one of these parameters is altered. The vehicle's public key $R_{pub}$ and message $m_{v,i}$. That's why the ECA-VFog is resistant to tweaks.

- Replay attack: The adversary executes this attack by reusing previously validated messages. Verifying the timeliness of the timestamp used in the parameters The message parameters $\{m_{v,i}, AID_v, R_{pub}, D_{v,i}, T_{V,i}, \sigma_{v,i}\}$, however, can thwart the attack. This means that the ECA-VFog is resilient against a replay assault.

- Man-in-the-middle attack: An attacker can deceive two vehicles or fog servers into thinking they are in constant contact with each other by exchanging information with them. It takes in private information and traffic messages, alters them, and relays the new versions to other vehicles or fog servers. Since all forms of mutual communication in the ECA-VFog require authentication, an unauthenticated attacker would be unable to launch such a campaign against a TRA. After registration, the vehicle contacts the KGC because the TRA has provided it with a partial pseudonym identity and private key, $PPID_V = Ps_v \oplus h_2(x \cdot Pub_{TRA})$. The ECA-VFog scheme would be safe from a MITM attack in this case.

## 5.2 Formal security verification using AVISPA tool

In order to formally validate the cryptographic protocol's security, we employ the Automated Validation of Internet Security Protocols and Applications (AVISPA) tool [35, 36]. To illustrate the security protocol, AVISPA makes use of the HLPSL [37, 38], which also enables us to state the security aspects of the protocol that need to be checked.

As can be seen in Fig 3 of the AVISPA architecture. SPAN is fed the protocol's CAS+ specification in Alice Bob notation, and it outputs a script in HLPSL. The HLPSL script is sent to an IF translator, which then runs it via the HLPSL to IF translator and the AVISPA backends for analysis. To ensure that the goals specified in HLPSL's goal section are met, AVISPA employs four backends: OFMC, CLAtSe, SATMC, and TA4SP. To ensure that the protocol is secure for the specified number of sessions or until an attack is discovered, the backend runs it through an infinite number of iterations. The HLPSL uses a state machine model of the protocol. There is a variable associated with each state, and as that variable's value shifts, the corresponding state shifts.

HLPSL uses the Doley-Yao threat model [39] to ensure that the cryptographic protocol is secure against man-in-the-middle assaults and replay assaults. The model assumes that the intruder can listen in on, steal, and forge any communication. Fig 4 show the simulated outcomes of the proposed ECA-VFog scheme. The protocol analysis tool ATSE found that out of 30 states, 24 are reachable, the translation time was 0.001 seconds, and the computation time was 0.001 seconds. With a depth of 6 heaps and a search period of 0.06 seconds, OFMC visits a total of 64 nodes. This result proves that the suggested protocol is secure against any kind of attack.

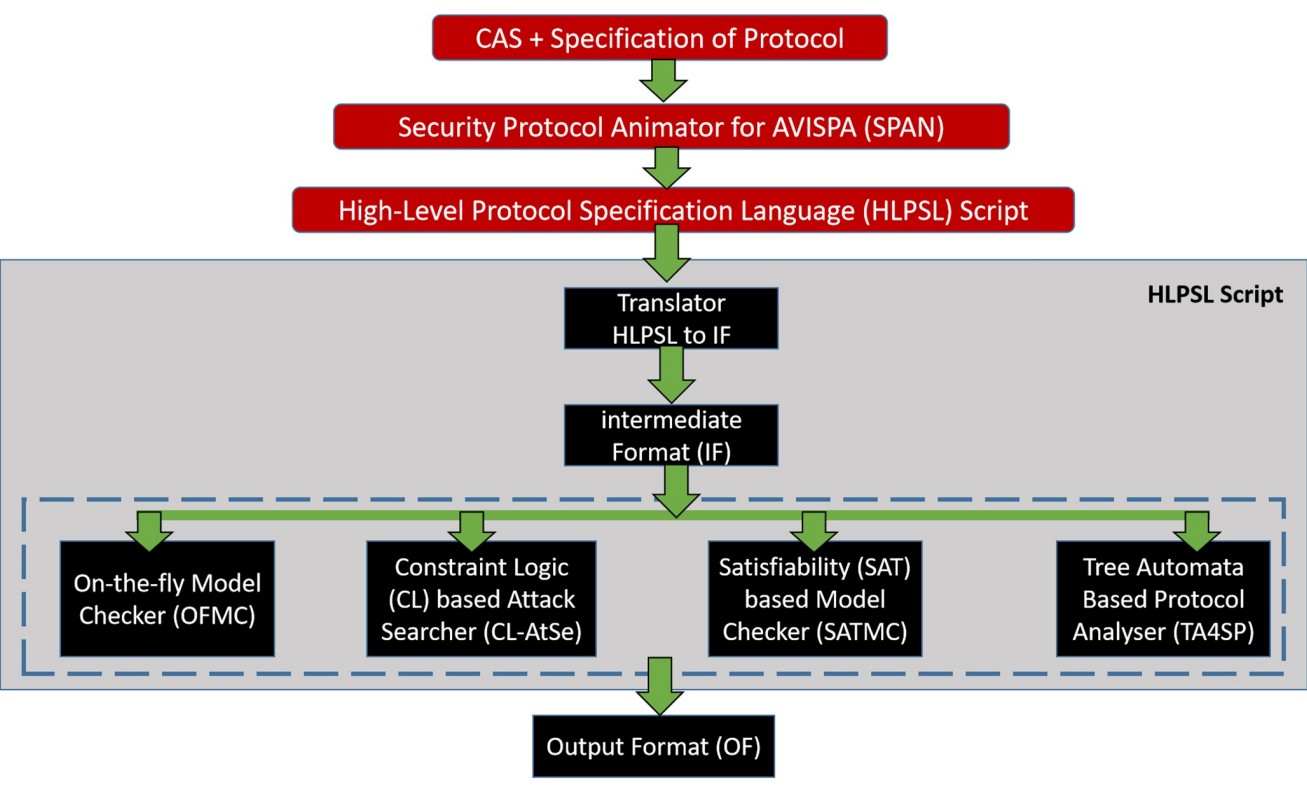

**Fig 3. Architecture model for AVISPA.**

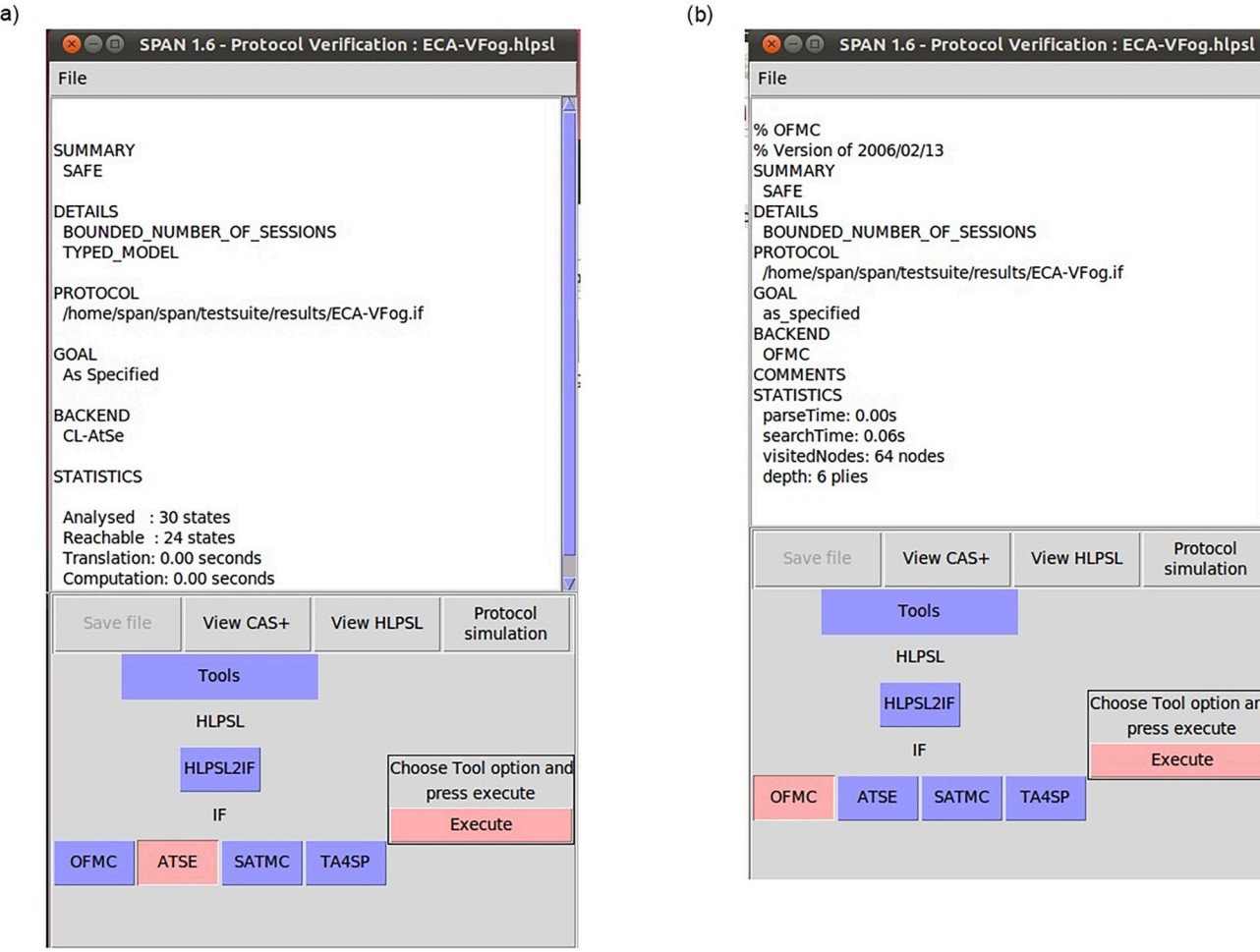

**Fig 4. AVISPA's results for our proposed ECA-VFog scheme.**

## 6 Performance evaluation

This section evaluates and compares the performance of the proposed ECA-VFog scheme with relevant schemes Bayat et al. [25], Wang et al. [26], Zhou et al. [32] and Liang et al. [33]. The evaluation criteria used for performance are computation, communication, and energy consumption overheads.

### 6.1 Evaluation and comparison of computation overhead

Here, we evaluate and compare the ECA-VFog scheme and some existing works Bayat et al. [25], Wang et al. [26], Zhou et al. [32] and Liang et al. [33] in terms of the overhead of computation. The following steps display the calculated achievement times (ms: millisecond) of cryptographic operations applied in the signing, verification, and batch verification of messages.

- $T_{bp}$: Operation of bilinear pairing (bp) in $Q, P \in G_1$. The running time for $T_{bp}$ is 6.101 ms.

- $T_{bp}^{sm}$: Operation of scalar multiplication $s \cdot P$ of the bp, $P \in G_1, s \in Z_q^*$. The running time for $T_{bp}^{sm}$ is 1.6765 ms.

**Table 2. Measurement of computation cost for authentication schemes.**

| Schemes | Message Signing (ms) | One Verification (ms) | Multiple Verification (ms) |
|---|---|---|---|
| Bayat et al. [25] | $6T_{bp}^{sm} + 1T_{bp}^{pa} + 1T_h \approx 10.0817$ | $3T_{bp} + 2T_{bp}^{sm} \approx 19.656$ | - |
| Wang et al. [26] | $T_{bp}^{pa} + 3T_{bp}^{sm} + T_h \approx 5.0522$ | $2T_{bp} + 3T_b^{pa}p + 2T_{bp}^{sm} + 2T_h \approx 15.6221$ | $2T_{bp} + 2nT_{bp}^{sm} + 3nT_{bp}^{pa} + 2nT_h \approx 12.202 + 3.4181n$ |
| Zhou et al. [32] | $2T_h + T_{ecc}^{sm} \approx 0.7849$ | $4T_{ecc}^{sm} + 3T_{ecc}^{pa} + 3T_h \approx 3.1997$ | $(2n+2)T_{ecc}^{sm} + 3nT_{ecc}^{pa} + 3nT_h \approx 1.57658 + 1.5814n$ |
| Liang et al. [33] | $6T_h + 4T_{ecc}^{sm} \approx 3.1376$ | $2T_{ecc}^{sm} + 4T_h + 4T_{bp} \approx 7.6708$ | $(3n+1)T_h + 2nT_{ecc}^{sm} + 4T_{bp} \approx 24.405 + 1.5688n$ |
| Our ECA-VFog | $1T_{ecc}^{pa} + 3T_{ecc}^{sm} + 1T_h \approx 2.3539$ | $2T_{ecc}^{pa} + 2T_{ecc}^{sm} + 1T_h \approx 1.5752$ | $(3n-1)T_{ecc}^{pa} + (n+1)T_{ecc}^{sm} + nT_h \approx 0.7965n + 0.7787$ |

- $T_{bp}^{pa}$: Operation of point addition $Q + P$ of the bp, $Q, P \in G_1$. The running time for $T_{bp}^{pa}$ is 0.0217 ms.

- $T_{ecc}^{sm}$: Scalar multiplication operation $s \cdot P$ of the ECC, $P \in G$, $s \in Z_q^*$. The running time for $T_{ecc}^{sm}$ is 0.7829 ms.

- $T_{ecc}^{pa}$: Point addition $P + Q$ of the bp, $Q, P \in G$. The running costs for $T_{ecc}^{pa}$ is 0.0042 ms.

- $T_h$: Function of general secure hash. The running time for $T_h$ is 0.001 ms.

This work utilizes the MIRACL cryptographic library [40] to time various cryptographic procedures. The machine runs Windows 10 on an Intel(R) Core(TM) i7-8550u processor at 1.80 GHz with 8 GB of RAM. Let's measure the computation overhead of the proposed ECA-VFog scheme. Table 2 displays the message singing, single verification, and batch verification computation costs for the proposed ECA-VFog scheme as well as other related works.

In the ECA-VFog, the computation of the message signing process needs one-point addition, three scalar multiplications, and two general secure hash functions. Therefore, the total running time of the message signing process is computed as $1T_{ecc}^{pa} + 3T_{ecc}^{sm} + 1T_h = 1 * 0.0042 + 3 * 0.7829 + 1 * 0.001 \approx 2.3539ms$. Computation of the single verification needs two-point additions, two scalar multiplications, and a general secure hash function, thus the entire computation time of this phase is computed as $2T_{ecc}^{pa} + 2T_{ecc}^{sm} + 1T_h = 2 * 0.0042 + 2 * 0.7829 + 1 * 0.001 \approx 1.5752ms$. Lastly, the computation of the batch verification needs (3n-1) addition point, (n + 1) scalar multiplication, and n general secure hash functions, thus the entire computation time of batch verification is computed as $(3n - 1)T_{ecc}^{pa} + (n + 1)T_{ecc}^{sm} + nT_h \approx 0.7965n + 0.7787ms$. Similarly, the computation costs of message singing, single verification, and batch verification are computed in terms of computation overhead in schemes Bayat et al. [25], Wang et al. [26], Zhou et al. [32] and Liang et al. [33]. Fig 5 lists a message signing and one verification for authentication schemes. Fig 6 shows multiple verifications of authentication schemes.

## 6.2 Evaluation and comparison of communication costs

Here, we evaluate and compare the ECA-VFog scheme and some existing works Bayat et al. [25], Wang et al. [26], Zhou et al. [32] and Liang et al. [33] in terms of the overhead of communication. To compare the ECA-VFog's communication overhead to those of other schemes, let's assume the following sizes of the various elements.

- $G_1$: The multiplicative cyclic group. The size of the item in $G_1$ is 128 bytes.

- $G$: The additional cyclic group. The size of the item in $G$ is 40 bytes.

- $Z_q^*$: The finite field. The size of the item in $Z_q^*$ is 20 bytes.

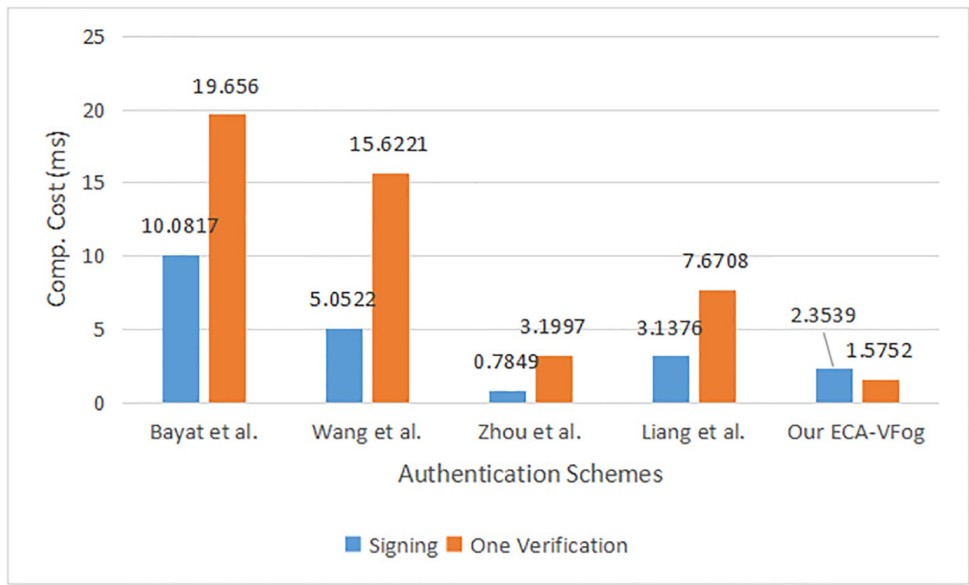

**Fig 5. Message signing and one verification for authentication schemes.**

- $T_i$: The timestamp. The size of the $Z_q^*$ is 4 bytes.

During the signing message of the proposed ECA-VFog scheme, the vehicle $v_i$ broadcasts the message parameters $\{m_{v,i}, AID_v, R_{pub}, D_{v,i}, T_{V,i}, \sigma_{v,i}\}$ to other vehicles and fog servers, where $\{R_{pub}, D_{v,i}\} \in G$, $\{AID_v, \sigma_{v,i}\} \in Z_q^*$, and timestamp $\{T_{V,i}\}$ = 4 bytes. Therefore, the bandwidth overhead of ECA-VFog is computed as 2 * 20 + 2 * 40 + 4 = 124 bytes. Likewise, the bandwidth overheads of Bayat et al. [25], Wang et al. [26], Zhou et al. [32] and Liang et al. [33]

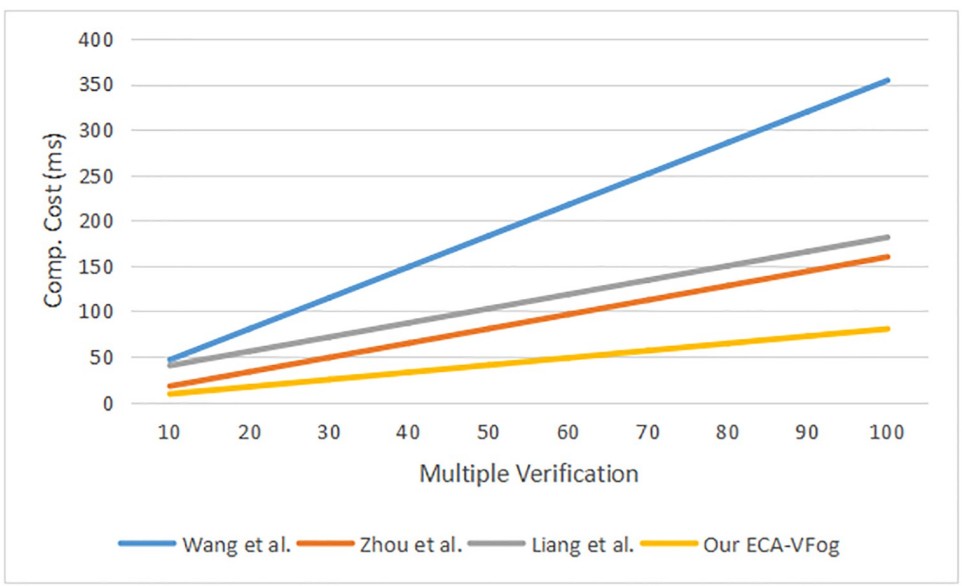

**Fig 6. Multiple verification for authentication schemes.**

**Table 3. Measurement of communication overhead for authentication schemes.**

| Schemes | Format | One Messages | N messages |
|---|---|---|---|
| Bayat et al. [25] | $\{r, T_{i1}, V, m, T_{i2}, PID_i, T_{i3}, ts_i\}$ | 772 bytes | 772*n bytes |
| Wang et al. [26] | $\{m_i, PID_{i,j}, TS_i, PK_i, V_i, W_i, U_i\}$ | 388 bytes | 388*n bytes |
| Zhou et al. [32] | $\{m_i, AID_i, Ts_i, r_i t_i, \sigma_i\}$ | 208 bytes | 208*n bytes |
| Liang et al. [33] | $\{PSUID_i^*, ID_b, pk_i, M_i, tt_i, \sigma_i\}$ | 256 bytes | 256*n bytes |
| Our ECA-VFog | $\{m_{v,i}, AID_v, R_{pub}, D_{v,i}, T_{V,i}, \sigma_{v,i}\}$ | 124 bytes | 124*n bytes |

are computed as 237, 388, 208, and 256 bytes, respectively. Table 3 provides a measurement of communication overheads.

## 6.3 Evaluation and comparison of energy consumption overhead

We use the procedure outlined in Table 2 to resolve the energy requirements of the ECA-VFog proposal. Using the full strength of the CPU (10.88 Watt) and the cost it takes to complete the task, the energy exhaustion can be computed as follows: E = P.t, whitch E is the power used, P is the full strength of the CPU, and t is the computation cost.

Let's calculate the energy exhaustion overhead of our ECA-VFog method. The energy exhaustion overhead $E$ of the ECA-VFog in message signing is computed as 10.88 * 2.3539 = 25.610432 mJ (E = P .t). Then, the energy exhaustion overhead E of ECA-VFog in single verification is computed as 10.88 * 1.5752 = 17.138176 mJ. Then, the power exhaustion overhead E of ECA-VFog for 10 messages (x = 10) in batch verification was calculated as 10.88 * 8.7437 = 95.131456 mJ. likewise, the energy exhaustion overhead of data signing, single authentication, and batch authentication is computed in schemes Bayat et al. [25], Wang et al. [26], Zhou et al. [32] and Liang et al. [33]. In Fig 7, we see a measurement among our ECA-V-Fog method and other existing schemes with regards to the message singing and verifying energy exhaustion overheads.

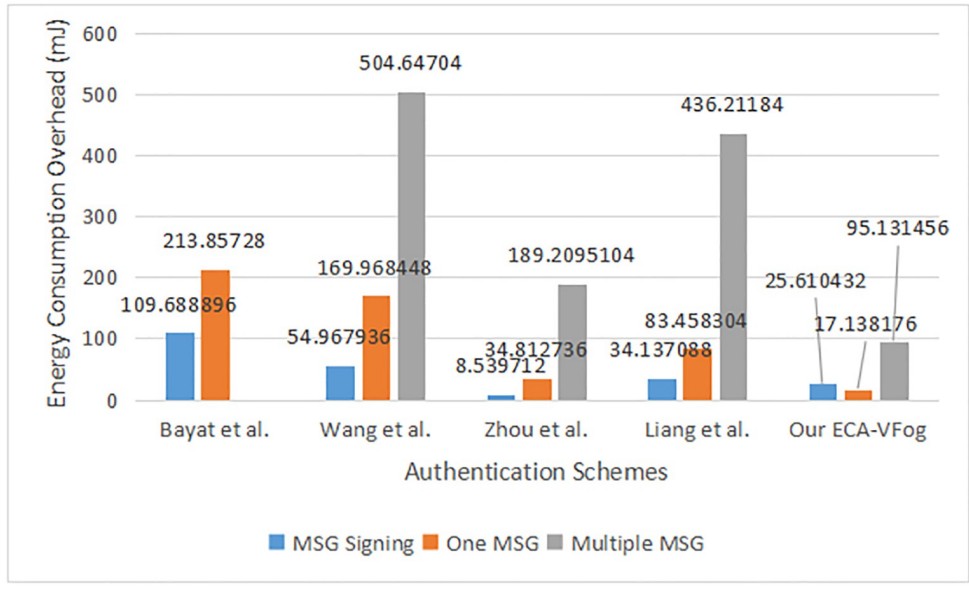

**Fig 7. Energy consumption overhead.**

## 7 Conclusion

This paper has proposed an efficient certificateless authentication with pseudonym and traceability called an ECA-VFog scheme for fog computing with a 5G-assisted vehicular system. The ECA-VFog scheme avoids complicated management of certificate and escrow of key in the public key infrastructure-assisted works the identity-assisted works, respectively. The proposed ECA-VFog scheme applied efficient operations based on elliptic curve cryptography that is supported by fog servers through 5G-BS. Security evaluation shows that the proposed ECA-VFog scheme satisfies security factors (data authenticity and integrity, pseudonym identity, traceability, unlinkability, location privacy, non-repudiation) and resists security attacks (Forgery, modified messages, replay, and man-in-the-middle) for vehicular fog computing based on 5G technology. The evaluation of the ECA-VFog scheme's performance shows that it is more efficient than relevant studies with regards to communication overhead, computation overhead, and energy consumption.

In future work, we extend this work to apply instead of 5G for secure compunctions in vehicular fog computing.

## Author Contributions

**Conceptualization:** Mahmood A. Al-Shareeda.

**Formal analysis:** Abdulwahab Ali Almazroi, Eman A. Aldhahri.

**Funding acquisition:** Abdulwahab Ali Almazroi, Eman A. Aldhahri, Selvakumar Manickam.

**Project administration:** Mahmood A. Al-Shareeda.

**Resources:** Abdulwahab Ali Almazroi, Selvakumar Manickam.

**Software:** Eman A. Aldhahri, Selvakumar Manickam.

**Visualization:** Eman A. Aldhahri.

**Writing – original draft:** Mahmood A. Al-Shareeda.

**Writing – review & editing:** Mahmood A. Al-Shareeda.

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
