## [Decision Letter · Decision Letter 0]

21 May 2023

PONE-D-23-13861ECA-VFog: An Efficient Certificateless Authentication Scheme for 5G-Assisted Vehicular Fog ComputingPLOS ONE

Dear Dr. Al-Shareeda‬‏,

Thank you for submitting your manuscript to PLOS ONE. After careful consideration, we feel that it has merit but does not fully meet PLOS ONE’s publication criteria as it currently stands. Therefore, we invite you to submit a revised version of the manuscript that addresses the points raised during the review process.

We look forward to receiving your revised manuscript.

Kind regards,

AbdulRahman A. ALsewari, Ph.D.

Academic Editor

PLOS ONE

Journal Requirements:

   "The authors extend their appreciation to the Deputyship for Research \\&  Innovation, Ministry of Education in Saudi Arabia for funding this research work through project number MoE-IF-UJ-22-04100409-X"

    "The authors extend their appreciation to the Deputyship for Research & Innovation, Ministry of Education in Saudi Arabia for funding this research work through project number MoE-IF-UJ-22-04100409-X"

   "The authors extend their appreciation to the Deputyship for Research \\&  Innovation, Ministry of Education in Saudi Arabia for funding this research work through project number MoE-IF-UJ-22-04100409-X"

Additional Editor Comments:

The paper proposts an efficient certificateless authentication scheme for 5G-assisted vehicular fog computing. I recommend the paper for acceptance for the following reasons. The structure of the paper is good. The paper is well written. The authors have done formal security analysis and informal security analysis. Moreover, the novelty presented in the paper is acceptable. However the paper required enhancement as suggested by the reviewer comments.

Reviewers' comments:

Reviewer's Responses to Questions

**Comments to the Author**

1. Is the manuscript technically sound, and do the data support the conclusions?

Reviewer #1: Yes

Reviewer #2: Yes

Reviewer #3: Yes

2. Has the statistical analysis been performed appropriately and rigorously? 

Reviewer #1: Yes

Reviewer #2: No

Reviewer #3: Yes

3. Have the authors made all data underlying the findings in their manuscript fully available?

Reviewer #1: Yes

Reviewer #2: Yes

Reviewer #3: Yes

4. Is the manuscript presented in an intelligible fashion and written in standard English?

Reviewer #1: No

Reviewer #2: No

Reviewer #3: Yes

5. Review Comments to the Author

Reviewer #1: In this work, the authors have proposed an efficient certificateless authentication scheme for 5G-assisted vehicular fog computing. I recommend the paper for acceptance for the following reasons.

The structure of the paper is good.

The paper is well written.

The authors have done formal security analysis and informal security analysis.

Moreover, the novelty of the paper is good.

I have found some minor problems in this paper which must be corrected in the revised version.

1) Section number must be given for all the sections and sub sections.

2) Why have the authors focused on 5G? Because nowadays many researches have started on 6G.

3) There are few typographical mistakes. For example, in the Introduction section, “this study is” must be “This study is”. In the same section, “Section reviews some related work” must be “Section 2 reviews some related work”. Modify the entire paragraph.

4) The authors must give expansion for all the abbreviations. For example check ECA.

5) Use unique symbols in the entire paper. For example, P is not italic in one place and it is italic in another place under the subheading Elliptic Curve Cryptography (ECC).

6) Related work section is weak. Because Some important and recent references are missing, the following references must be totally added in the Section "References" (otherwise, the reference is not enough, then it must be revised again until it is enough):

Efficient certificateless designated verifier proxy signature scheme using UAV network for sustainable smart city

Efficient certificateless conditional privacy-preserving authentication for VANETs

Secure and Efficient Authenticated Key Management Scheme for UAV-Assisted Infrastructure-Less IoVs

RAKI: A Robust ECC Based Three-party Authentication and Key Agreement Scheme for Medical IoT

A robust mutual and batch authentication scheme based on ECC for online learning in Industry 4.0

Reviewer #2: Aiming at the security and privacy requirements of 5g assisted vehicle network, this paper proposes a certificateless authentication scheme based on ECC under the vehicular fog computing architecture. In addition, security analysis and experimental evaluation are carried out. However, as a whole, this paper is not innovative enough. In addition, there are the following problems.

1. There are many errors in the abbreviation format of words in the article, for example, the full name of VANET is not explained for the first time. In addition, the full names of PKI and ID also appear several times. Therefore, it is necessary to correct the abbreviation errors that appear in the paper as a whole.

2. The literature annotation in this paper is quite arbitrary. For example, references [1-11] are not very relevant to the tagged sentences. However, in the third and fourth paragraphs of the introduction, there is a lack of literature sources when presenting related work. Therefore, it is very necessary to carefully review the appropriateness of each cited paper. In addition, the missing references need to be supplemented.

3. In the introduction section, the main contributions of the paper are introduced without elaborating on the innovative ideas of the paper. In particular, in the main contribution section, the differences with existing work are not reflected, and the innovative work of the paper is not described.

4. In the related work, the author lists the current work, but the analysis of its shortcomings is too few. In addition, there are few related works on certificateless authentication schemes that are most relevant to the proposed scheme, and only three papers cannot fully reflect the current research progress.

5. In the system model section, TRA,KGC,FS and 5G-BS are assumed to be fully trusted, which is obviously completely impossible in the real scenario. Thus, it makes sense to at least assume that the third party is semi-trusted.

6. In the section of the ECA-VFog scheme, there is no detailed innovative explanation. Neither based on innovation theory nor using new technologies.

7. It is suggested to prove the security of the proposed scheme in the standard model or the random oracle model.

8. In the whole article, the integration of 5G technology and Internet of vehicles is not reflected.

Therefore, although the performance evaluation of the experiments is good, the effectiveness of the proposed scheme needs to be considered in more realistic attack scenarios.

Reviewer #3: This manuscript has proposed an efficient certificateless authentication called the ECA-VFog scheme for fog computing with 5G-assisted vehicular systems by applying efficient operations based on elliptic curve cryptography that is supported by a fog server through a 5G-base station. The topic of manuscript is really interested and timed. The paper is written well, there are research voice in these sections. However, the paper required the following comments and concerns to improve the quality of the presentation.

1-In abstract part, the outcome (in numerical) of this proposed have been missing, please add them.

2-In related work, the paper strongly required to add some relevant researches such as (An efficient conditional privacy-preserving authentication scheme for the prevention of side-channel attacks in vehicular ad hoc networks)

3- What are the benefit from 5G and fog computing? This needs to be clarified in the paper

4- page 2, line 50-55 >> double checking, there is missing number/name of sections

5- Table 2, need to align the text

6- Please add future work in conclusion part

6. PLOS authors have the option to publish the peer review history of their article (what does this mean?). If published, this will include your full peer review and any attached files.

Reviewer #1: No

Reviewer #2: No

Reviewer #3: **Yes**

---

## [Author Response · Author response to Decision Letter 0]

25 May 2023

Original Manuscript ID: PONE-D-23-13861

Original Article Title: “ECA-VFog: An Efficient Certificateless Authentication Scheme for 5G-Assisted Vehicular Fog Computing”

To: PLOS ONE

Re: Response to reviewers

Dear Editor,

Thank you for allowing a resubmission of our manuscript, with an opportunity to address the reviewers’ comments.

Best regards,

Dr. Mahmood A. Al-Shareeeda.

Reviewer 1: Concern #1: Section number must be given for all the sections and sub-sections.

Reviewer 1: Response #1: Thanks for the valuable comments. We revise the manuscript by adding sections and sub-sections as follows.

Reviewer 1: Concern #2: Why have the authors focused on 5G? Because nowadays many researcher have started on 6G.

Reviewer 1: Response #2: Thanks for the valuable comments. The 5G is the scope of this research. In future work, we extend this work to apply 6G instead of 5G for secure compunctions. The revised sentence has been added in conclusion section of the paper as follows.

Reviewer 1: Concern #3: There are few typographical mistakes. For example, in the Introduction section, “this study is” must be “This study is”. In the same section, “Section reviews some related work” must be “Section 2 reviews some related work”. Modify the entire paragraph.

Reviewer 1: Response #3: Thanks for the valuable comments. We revise the manuscript by modifying the entire paragraph as follows.

Reviewer 1: Concern #4:  The authors must give expansion for all the abbreviations. For example check ECA.

Reviewer 1: Response #4: Thanks for the valuable comments. We revise the manuscript by adding missing definition of abbreviations in the following table.

Reviewer 1: Concern #5:  Use unique symbols in the entire paper. For example, P is not italic in one place and it is italic in another place under the subheading Elliptic Curve Cryptography (ECC).

Reviewer 1: Response #5: Thanks for the valuable comments. We revise the manuscript by remodify entire subheading Elliptic Curve Cryptography (ECC) as follows.

Reviewer 1: Concern #6:  Related work section is weak. Because Some important and recent references are missing, the following references must be totally added in the Section "References" (otherwise, the reference is not enough, then it must be revised again until it is enough):

Reviewer 1: Response #6: Thanks for the valuable comments. We revise the manuscript by reviewing the relevant articles as follows. 

Xu et al. [27] constructed certificateless designated verifier proxy signature using unmanned aerial vehicles (UAVs) to address privacy and security concerns in smart city systems. Ming et al. [28] designed an efficient certificateless authentication scheme by achieving a security-enhanced solution and addressing massive communication overhead, security vulnerability, and computational complexity. Tan et al. [29] proposed a certificateless group authentication scheme based on UAV in order to achieve security communication in infrastructure-less internet of vehicle (IoV). Zhou et al. [30] designed a secure ECC scheme by using key agreement and a three-party authentication scheme in medical IoT. Rajasekaran et al. [31] proposed a secure ECC method that supports batch verification and mutual authentication for online learning in Industry 4.0.

Reviewer 2: Concern #1:  There are many errors in the abbreviation format of words in the article, for example, the full name of VANET is not explained for the first time. In addition, the full names of PKI and ID also appear several times. Therefore, it is necessary to correct the abbreviation errors that appear in the paper as a whole.

Reviewer 2: Response #1: Thanks for the valuable comments. We revise the manuscript by defining the abbreviation in Table and introduction section as it first appears.

Reviewer 2: Concern #2:  The literature annotation in this paper is quite arbitrary. For example, references [1-11] are not very relevant to the tagged sentences. However, in the third and fourth paragraphs of the introduction, there is a lack of literature sources when presenting related work. Therefore, it is very necessary to carefully review the appropriateness of each cited paper. In addition, the missing references need to be supplemented.

Reviewer 2: Response #2: Thanks for the valuable comments. In this manuscript, we really focus on updated references that are relevant to our work. We revise the manuscript by adding more references based on suggested reviewers as follows.

Xu et al. [27] constructed certificateless designated verifier proxy signature using unmanned aerial vehicles (UAVs) to address privacy and security concerns in smart city systems. Ming et al. [28] designed an efficient certificateless authentication scheme by achieving a security-enhanced solution and addressing massive communication overhead, security vulnerability, and computational complexity. Tan et al. [29] proposed a certificateless group authentication scheme based on UAV in order to achieve security communication in infrastructure-less internet of vehicle (IoV). Zhou et al. [30] designed a secure ECC scheme by using key agreement and a three-party authentication scheme in medical IoT. Rajasekaran et al. [31] proposed a secure ECC method that supports batch verification and mutual authentication for online learning in Industry 4.0.

Reviewer 2: Concern #3:  In the introduction section, the main contributions of the paper are introduced without elaborating on the innovative ideas of the paper. In particular, in the main contribution section, the differences with existing work are not reflected, and the innovative work of the paper is not described.

Reviewer 2: Response #3: Thanks for the valuable comments. Unlike the related work, this paper is based on the fog computing and the 5G technology to trad-off of between security and performance. Hence, we revise the manuscript by adding the innovative of the proposal as follows.

Reviewer 2: Concern #4:  In the related work, the author lists the current work, but the analysis of its shortcomings is too few. In addition, there are few related works on certificateless authentication schemes that are most relevant to the proposed scheme, and only three papers cannot fully reflect the current research progress.

Reviewer 2: Response #4: Thanks for the valuable comments. We revise the manuscript by adding some relevant certificatesless-based schemes. Seven relevant researches have been reviewed in this subsection as follows.

Reviewer 2: Concern #5:   In the system model section, TRA,KGC,FS and 5G-BS are assumed to be fully trusted, which is obviously completely impossible in the real scenario. Thus, it makes sense to at least assume that the third party is semi-trusted.

Reviewer 2: Response #5: Thanks for the valuable comments. We revise the manuscript by adding the assumption of each unit in the architecture model as follows.

Reviewer 2: Concern #6:  In the section of the ECA-VFog scheme, there is no detailed innovative explanation. Neither based on innovation theory nor using new technologies.

Reviewer 2: Response #6: Thanks for the valuable comments. We revise the manuscript by adding innovation explanation (unlike the related work) as follows.

Reviewer 2: Concern #7: It is suggested to prove the security of the proposed scheme in the standard model or the random oracle model.

Reviewer 2: Response #7: Thanks for the valuable comments. In the related work, the prove of the security based on formal and informal analysis. Therefore, this paper uses the AVISPA simulator tool as standard model to show security attacks resistances.

Reviewer 2: Concern #8:   In the whole article, the integration of 5G technology and Internet of vehicles is not reflected.

Reviewer 2: Response #8: Thanks for the valuable comments. 5G doesn’t compute and save any the security parameters for vehicles. The main aim of 5G is not only to serve a large number of vehicles within its communication range, it also forwards messages among vehicles and fog servers.

Reviewer 3: Concern #1:  In abstract part, the outcome (in numerical) of this proposed have been missing, please add them.

Reviewer 3: Response #1: Thanks for the valuable comments. We revise the manuscript by adding the outcome (in numerical) of this proposed in abstract section.

Reviewer 3: Concern #2:  In related work, the paper strongly required to add some relevant researches such as (An efficient conditional privacy-preserving authentication scheme for the prevention of side-channel attacks in vehicular ad hoc networks)

Reviewer 3: Response #2: Thanks for the valuable comments. We revise the manuscript by reviewing this reference as follows.

Reviewer 3: Concern #3:  What are the benefit from 5G and fog computing? This needs to be clarified in the paper

Reviewer 3: Response #3: Thanks for the valuable comments. We revise the manuscript by adding the benefit from fog computing and 5G in system model as follows.

-5G-Base Station (5G-BS): The 5G-BSs are stationary base stations set up by the side of the road. Its only use is as a bridge between vehicles, fog servers, and TRA, and it lacks both computing and storage capabilities. This is because it can accommodate a wide variety of device-to-device (D2D) communication standards. Because 5G-BSs are hardware, they are immune to attacks.

-Fog Server: Fog server is the roadside infrastructure that enables vehicle-to-infrastructure (V2I) communication and can also realize inter-infrastructure (I2I) communication. FSs can simultaneously relay multiple messages collected from vehicles. FSs are stationed in various areas behind 5G-BS, and passing vehicles are made aware of their location. By sharing information, it can also boost circulation in the area covered by 5G-BS. 

Reviewer 3: Concern #4:  page 2, line 50-55 >> double checking, there is missing number/name of sections

Reviewer 3: Response #4: Thanks for the valuable comments. We revise the manuscript by adding the missing number as follows.

Reviewer 3: Concern #5:   Table 2, need to align the text

Reviewer 3: Response #5: Thanks for the valuable comments. We revise the manuscript by aligning the table to text of the paper as follows.

Reviewer 3: Concern #6:  Please add future work in conclusion part

Reviewer 3: Response #6: Thanks for the valuable comments. We revise the manuscript by adding future works as follows.

---

## [Decision Letter · Decision Letter 1]

5 Jun 2023

ECA-VFog: An Efficient Certificateless Authentication Scheme for 5G-Assisted Vehicular Fog Computing

PONE-D-23-13861R1

Dear Dr. Al-Shareeda‬‏,

We’re pleased to inform you that your manuscript has been judged scientifically suitable for publication and will be formally accepted for publication once it meets all outstanding technical requirements.

Kind regards,

AbdulRahman A. ALsewari, Ph.D.

Academic Editor

PLOS ONE

Additional Editor Comments (optional):

Reviewers' comments:

Reviewer's Responses to Questions

**Comments to the Author**

1. If the authors have adequately addressed your comments raised in a previous round of review and you feel that this manuscript is now acceptable for publication, you may indicate that here to bypass the “Comments to the Author” section, enter your conflict of interest statement in the “Confidential to Editor” section, and submit your "Accept" recommendation.

Reviewer #1: All comments have been addressed

Reviewer #3: All comments have been addressed

2. Is the manuscript technically sound, and do the data support the conclusions?

Reviewer #1: Yes

Reviewer #3: Yes

3. Has the statistical analysis been performed appropriately and rigorously? 

Reviewer #1: Yes

Reviewer #3: Yes

4. Have the authors made all data underlying the findings in their manuscript fully available?

Reviewer #1: Yes

Reviewer #3: Yes

5. Is the manuscript presented in an intelligible fashion and written in standard English?

Reviewer #1: Yes

Reviewer #3: Yes

6. Review Comments to the Author

Reviewer #1: The authors have done all the corrections given by me in the previous round and hence the paper can be accepted in the present format.

Reviewer #3: The authors have addressed all the comments. I recommend to accept the paper in its current format.

7. PLOS authors have the option to publish the peer review history of their article (what does this mean?). If published, this will include your full peer review and any attached files.

Reviewer #1: No

Reviewer #3: No

---

## [Editor Report · Acceptance letter]

14 Jun 2023

PONE-D-23-13861R1 

ECA-VFog: An Efficient Certificateless Authentication Scheme for 5G-Assisted Vehicular Fog Computing 

Dear Dr. Al-Shareeda‬‏:

I'm pleased to inform you that your manuscript has been deemed suitable for publication in PLOS ONE. Congratulations! Your manuscript is now with our production department. 

Kind regards, 

on behalf of

Dr. AbdulRahman A. ALsewari 

Academic Editor

PLOS ONE